# Autoimmunity, COVID-19 Omicron Variant, and Olfactory Dysfunction: A Literature Review

**DOI:** 10.3390/diagnostics13040641

**Published:** 2023-02-09

**Authors:** Yonatan Shneor Patt, Lior Fisher, Paula David, Moriah Bergwerk, Yehuda Shoenfeld

**Affiliations:** 1Zabludowicz Center for Autoimmune Diseases, Sheba Medical Center, Tel Hashomer, Ramat Gan 5265601, Israel; 2Department of Medicine ‘B’, Sheba Medical Center, Tel Hashomer, Ramat Gan 5262100, Israel; 3Sackler Faculty of Medicine, Tel-Aviv University, Tel Aviv 6209813, Israel; 4Laboratory of the Mosaic of Autoimmunity, Saint Petersburg State University, 199034 Saint-Petersburg, Russia

**Keywords:** SARS-CoV-2, COVID-19, Omicron variant, smell, olfactory dysfunction, anosmia, autoimmunity, systemic lupus erythematosus, transmembrane serine protease 2 (TMPRSS2)

## Abstract

Smelling is a critical sense utilized daily. Consequently, smelling impairment or anosmia may lead to a reduction in life quality. Systemic diseases and particular autoimmune conditions can impair olfactory function; among others are Systemic Lupus Erythematosus, Sjögren Syndrome, and Rheumatoid Arthritis. Interactions between the olfactory process and the immune systems cause this phenomenon. Alongside autoimmune conditions, in the recent COVID-19 pandemic, anosmia was also described as a prevalent infection symptom. Nevertheless, the occurrence of anosmia is significantly less common in Omicron-infected patients. Several theories have been proposed to explain this phenomenon. One possibility is that the Omicron variant preferentially enters host cells via endocytosis, rather than plasma cell membrane fusion. This endosomal pathway is less dependent on the activation of Transmembrane serine protease 2 (TMPRSS2), expressed at the olfactory epithelium. As a result, the Omicron variant may have reduced efficiency in penetrating the olfactory epithelium, leading to a lower prevalence of anosmia. Furthermore, olfactory changes are known to be associated with inflammatory conditions. The Omicron variant elicits a less robust autoimmune and inflammatory response, believed to reduce the probability of anosmia. This review elaborates on the commonalities and differences in autoimmune and COVID-19 omicron-associated anosmia.

## 1. Introduction

Smelling or olfaction, the ability to perceive odor or scent, is a fundamental sensory modality imperative in detecting hazards, beneficial nutrients, pheromones, and considerably influences on the ability to taste.

The olfaction process is facilitated by the olfactory receptors in the olfactory epithelium of the nasal cavity of vertebrates. Sensory processing is generated by combining electrophysiological and biochemical processes that translate molecular data into sensation and enables a highly specific ability to distinguish and detect numerous low molecular mass compounds (odorants) [1].

The olfactory mucosa is composed of the olfactory epithelium and the lamina propria. The olfactory epithelium contains various cell types, including olfactory sensory neurons, sustentacular cells, and basal stem cells. Olfactory sensory neurons, responsible for the sense of smell, are equipped with specialized cilia that detect odorant molecules. Sustentacular cells provide structural support and play a role in phagocytosis [2], the transformation of odorants, and the production of metabolic enzymes [3]. Basal cells, on the other hand, function as stem cells capable of generating new sensory neurons and sustentacular cells throughout an individual’s lifetime [4].

The cribriform plate is a perforated segment of the ethmoid bone, separating the frontal cerebrum lobe and the nasal cavity. The olfaction process initiates by odorant molecules entering the nasal passages and advancing to the roof of the nasal cavity to the cribriform plate. When in the nasal cavity, the odorant molecules interact with receptors located on the primary cilia of olfactory neurons. A single olfactory neuron expresses a specific structure of protein receptor on its dendritic branches. However, a particular odorant can bind to various protein-binding receptors [5].

When engaged, the olfactory neurons generate an electrical impulse transmitted to the neural synapses junctions of the olfactory bulb structure and processed. Additional processing of the olfactory stimulant is performed in higher cortical regions. The orbitofrontal cortex (OFC), the amygdala, and the hippocampus take part in the translation of the olfactory stimulant and incorporation in the thought processes, perception of emotional reactions, forming memory and learning process. The multiregional involvement of the olfactory stimuli, consisting of the amygdala, pre-pyriform cortex, entorhinal cortex, and hippocampus, is perceived to account for the relationship between smell, emotions, and memory formation [6].

Inability or impairment of the olfactory process can lead to markedly reduced quality of life. The lack of enjoyment from eating and drinking can be associated with alterations in dietary and social patterns and behavior. A wide array of olfactory impairments is linked with pathophysiologic processes of head and brain trauma, elderly age, and toxic exposure. Along with the mechanical, toxic, and age-related mechanisms, the systemic, immune-mediated pathophysiology of olfactory impairment has been subject to interest in recent years. Neuroimmunological mechanisms can target the olfactory system and process, adding to the complexity of clinical diagnosis and management of systematic immune-mediated diseases [7].

Olfactory impairment was also often observed among COVID-19-infected patients. The damaging involvement of the smelling sense is a unique phenomenon related to the disease and is even considered a diagnostic tool for identifying infected individuals.

Our study aimed to evaluate the inflammatory process in anosmia in several systemic disease, in particular autoimmune and autoinflammatory conditions, in addition to the examination of the recently suggesting mechanisms and updates regarding the COVID-19-induced olfactory dysfunction (OD).

## 2. Olfactory Dysfunction and Autoimmunity

There are parallels between the immune and olfactory systems, both of which respond to stimuli in our surroundings. An individual’s sense of smell is impacted by a mix of genetics, hormones, and environmental factors, much such as the interplay of factors that govern autoimmunity [8].

In addition to the physiological similarity, various interactions between the olfactory process and the immune systems have been recognized and described. In mice models, Kivity et al. [9] demonstrated odorants suppression of immunological reaction. Inhaled odorants were linked to the suppression of mast cell activation by inhibiting the stress-induced activation of suppressor T lymphocytes.

In accordance with the observed association between autoimmunity and smelling alteration, several autoimmune conditions were evaluated for hyposmia (Table 1).

### 2.1. Olfactory Dysfunction among SLE Patients

Central nervous system involvement in systemic lupus erythematosus (SLE), referred to as neuropsychiatric systemic lupus erythematosus (NPSLE), is highly prevalent among SLE patients. Published in 1999 by the American College of Rheumatology, is a proposed standardized nomenclature for the definitions of 19 neuropsychiatric expression of SLE [10].

Pathogenesis of CNS involvement in SLE patients has been attributed to autoantibody-mediated neuronal dysregulation, vascular damage, and coagulation pathway imbalance. Multiple autoantibodies have been studied and associated with SLE neuropsychiatric expression, including antineuronal, anti-P ribosomal, antiphospholipid, anti-endothelial, antihuman N-methyl-d-aspartate receptor, and anti-Nedd5 C-ter antibodies [11].

The first reported antibody association between psychosis and neuropsychiatric lupus was the description of elevated anti-P ribosomal antibodies in such patients; the positive correlation to CNS involvement was a significant novelty [12].

Katzav et al. [13] aimed to evaluate and describe the neurological impairment of anti-P ribosomal antibodies mitigated damage and SLE. To accomplish this aim, anti-P ribosomal antibodies were injected into the brain’s ventricles of a rodent model of SLE. Affected mice demonstrated behavior previously linked to depression and impaired olfaction detection utilizing a smell threshold test. Furthermore, monoclonal antibodies therapy aimed at the anti-P ribosomal antibodies and antidepressant drugs or aromatherapy was found to improve depressive-like behavior. In addition to this finding, specific SLE autoantibody histological staining resulted in apparent olfactory-related brain structures [14], further strengthening the association of olfactory and neurological autoantibody pathology.

Shoenfeld et al. [15] assessed the Olfactory functions of 50 SLE patients and 50 matched controls in a different study aimed at understanding olfactory impairment in SLE. A Sniffin’ Sticks test was performed using three stages: threshold levels, discrimination levels, and identification levels of various odors. Comparing the groups revealed a significant decrease in smell abilities in the SLE patient group, as 46% were hyposmic compared with 25% in the control group, and 10% were anosmic relative to none in the controls. The quantitative effect of olfactory impairment in the study was found to correlate with SLE disease activity and other CNS manifestations. Similar findings were found by Chen et al. [16] when the olfactory function level of 65 patients with SLE and 50 healthy patients participants was measured using the Connecticut Chemosensory Clinical Research Center (CCCRC) method. OD was demonstrated in active SLE patients and correlated with SLE disease activity and anti-ribosomal P protein antibody levels.

In an additional study, a comparison of 143 SLE patients, 57 systemic sclerosis (SSc) patients, and 166 healthy volunteers of olfactory function was performed using the Sniffin’ Sticks test to detect OD and magnetic resonance imaging to evaluate the amygdalae and hippocampi volumes. In this study, Olfactory impairment was seen in 78 (54.5%) SLE patients, 35 (59.3%) SSc patients, and 24 (14.45%) healthy controls (*p* < 0.001). Olfactory assessment scoring was consistently low in all three assessment phases in the SLE and SSc patients compared with the control group. Anti-ribosomal P antibodies concentrations, older age, disease activity, depression symptoms, and smaller left hippocampus volume were all associated with OD [17].

### 2.2. Olfactory Dysfunction among Sjögren Syndrome (SjS) Patients

Sjögren syndrome is an inflammatory autoimmune rheumatic disorder characterized by a dysregulated and autoantibodies-mediated infiltration and destruction of exocrine glands [18]. In 1972, Heinkin et al. [19] were among the first to publish impaired olfaction associated with Sjögren syndrome in 29 patients.

In a later study, Kamel et al. [20] assembled 28 SjS patients and 37 healthy control patients and measured olfaction functioning. In this comparison, a reduction in the olfaction ability and taste threshold was prominent in the SjS group. Additionally, nearly half of the SjS group presented with partial or complete loss of the olfaction ability.

Multiple processes were previously noted as potential contributors to the olfaction deficits among SjS patients, including mucosal dryness, decreased odorant carrier, glycosylated proteins (mucin) production by epithelial tissues, and recurrent or chronic or chronic rhinosinusitis [20].

### 2.3. Olfactory Dysfunction among Rheumatoid Arthritis Patients

Rheumatoid arthritis (RA) is an inflammatory, autoimmune polyarthritis of unknown cause. It may lead to deformity and physical disability through progressive joint, cartilage, and bone erosions [21]. Along with the characteristic musculoskeletal system manifestations, it has been previously reported that RA-diagnosed patients may suffer from olfactory impairment. A study by Steinbach et al. [22] suggested that RA patients may suffer from impaired olfactory function, a novel clinical observation of RA disease manifestation. The authors hypothesized that systemic inflammation might affect the intra-nasal mucosa in a similar fashion to the articular synovial damage.

### 2.4. Olfactory Dysfunction among Autoimmune Thyroid Disease Patients

In 1975, the first report describing olfactory impairment as associated with primary hypothyroidism was published [23]. In this study, 39% of patients reported alteration of some degree in their olfaction ability. Considering that these patients did not inform smelling-related symptoms before the diagnosis of hypothyroidism, along with an observed improvement after 12 weeks of thyroxine replacement therapy, it has been hypothesized that thyroid hormones influence and modulate olfactory sensitivity.

In a mice-based animal model of reduced thyroid function, a significant decrease in olfactory function was observed. This observation was reversed when subject mice were provided supplemental thyroxine in the drinking water. These findings may indicate that hypothyroidism may influence and impair olfactory function, and thyroxine supplementation may restore olfactory function in mice similarly to humans [24].

### 2.5. Olfactory Dysfunction among Systemic Sclerosis Patients

Systemic sclerosis (SSc) is an immune-mediated, chronic multisystem disease characterized by connective tissue damage, vascular dysfunction, and internal organ fibrosis [25].

In a large cohort study aimed to assess olfaction ability, 143 SLE patients, 57 SSc patients, and 166 healthy volunteers were recruited. The SLE and the SSc patient groups had markedly reduced olfactory assessment scores compared with the healthy volunteers’ group. When revising the SSc patients, olfactory impairment was associated with elderly age, disease severity, smaller hippocampi volumes, and smaller right amygdala volume [17].

## 3. Olfactory Dysfunction and COVID-19

The coronavirus disease 2019 (COVID-19), brought on by the severe acute respiratory syndrome coronavirus 2 (SARS-CoV-2), was first discovered in Wuhan, China, in December 2019 [26]. On 11 March 2020, the World Health Organization declared the COVID-19 a worldwide pandemic after the virus began to spread globally. With more than 6.8 million fatalities, the epidemic is still harmful, despite significant international efforts to develop and distribute a viable disease vaccine [27].

Despite the successful development of vaccinations, viral genome adaptive changes are a prominent reason for the COVID-19 pandemic not being fully controlled. Due to the virus’s multiple strains and heterogeneity, reinfection and infection of vaccinated individuals is prevalent [28]. The viral genetic alterations may result in different transmissibility patterns, symptomatology, and pathogenicity.

First reported in South Africa in November 2021 [29], the SARS-CoV-2 Omicron (B.1.1.529) strain is a cause of global concern. In a literature analysis, Mohsin et al. [30] presented multiple studies suggesting that the Omicron variation spread faster than other variants. The fast transmissibility of the Omicron variant was also demonstrated in partly and fully vaccinated patients [31]. Although presumed more contagious, the Omicron variant was also found to be less dangerous and is associated with a reduced risk of hospital admissions, ICU hospitalizations, need for mechanical ventilation, hospital stays, and overall mortality [30,32].

Among common symptoms, including myalgia, cough, fever, and other respiratory abnormalities associated with COVID-19, a notable sign reported in many COVID-19 variations is OD [33].

A meta-analysis of 23,533 COVID-19 patients found that 38% presented OD [34]. In an additional study conducted in Iran, the incidence of OD among COVID-19-affected patients was significantly higher, with 76.24% reporting a rapid onset of OD and 60.90% reporting this symptom as permanent [35]. It is also noteworthy that many patients describe anosmia as the only symptom experienced during illness or the presenting symptom of infection [36]. Considering the prevalence and incidence shown in earlier studies, efforts are made to utilize OD as a diagnostic marker to distinguish and diagnose infected individuals based on olfactory alterations [33,37].

Interestingly, the Omicron variant has been reported to have a lower OD frequency. Collecting data from 12 reports of 190,778 affected people, Butowt et al. [38] generated a pooled weighted value of olfactory impairment of 13%. In another study, a comparison was made between 338 Omicron variant-infected participants and 441 patients were infected during the control period while other strains were more prevalent. The prevalence of self-reported OD in the Omicron group was substantially lower with 32.5% versus 66.9% in the comparable control group [39].

A retrospective study from Brazil included 6053 Omicron variant mild COVID-19 cases and compared OD incidence between the Omicron variant-affected cohort and between different variants, including original lineages (B.1.1.28 and B.1.1.33), Gamma, and Delta. It was found that the Omicron phase cohort had a reduced incidence of OD compared with those who acquired it during the original lineage era (5.8% vs. 52.6%, respectively, adjusted OR 0.07, PV 0.001). Even after further correcting for immunization status, OD occurred less often during the Omicron era than during the Gamma period [40].

In an additional study by Menni et al. [41], smelling alteration rates were measured in patients with COVID-19 infections throughout two time periods, with two different variants predominating: the Delta and the Omicron variants. Patients infected during the Omicron dominant phase had substantially less anosmia than those infected during the Delta dominant phase (16.7% vs. 52.7%, respectively; OR 0.17, 95% CI 0.16–0.19, *p <* 0.001). Additional reports have been published concerning the Omicron variant’s decreased incidence of OD, corroborating earlier studies’ results [42,43].

### 3.1. Mechanisms of Olfactory Dysfunction

Although the focus of multiple studies, the exact mechanism of COVID-19-induced anosmia remains unknown. Olfactory disorders are often characterized and divided as either sensorineural or conductive. Losing the olfactory receptor neurons and other sensory structures defines sensorineural impairments. Conductive deficiencies, such as rhinosinusitis and nasal polyps, may block the passage of inspired odorants to the olfactory epithelium in the nasal cavity [44]. Given that nasal congestion, causing blocking of odorant passage is considered uncommon in COVID-19 infection, conductive loss in individuals infected with the virus does not significantly contribute to their anosmia [39,45].

In a literature review, Meng et al. [46,47] reviewed several dominant hypotheses for the pathophysiology of COVID-19-induced anosmia. Angiotensin-converting enzyme 2 (ACE2) receptor and transmembrane serine protease 2 (TMPRSS2) have both been implicated in the invasion and fusion of the SARS-CoV-2 virion with host cells [48]. It can be inferred that cells with a high density of these receptors are more vulnerable to viral infection.

Brann et al. [49] reported sustentacular or epithelial basal cells to contain ACE2 receptors more prominently than the olfactory sensory neurons. Considering this finding, it may be presumed that sustentacular cell destruction may result in reduced olfactory function even if the virus does not reach the sensory neurons. The olfactory modulation in the nasal epithelial cells can be attributed to SARS-CoV-2 virions damage of sustentacular cells, either in their role in detoxifying airborne toxins, their ability to produce metabolites, or their involvement in the signal transduction of fragrance [50]. Hamster models are a source of valuable data indicating viral accumulation solely in sustentacular cells [51]. Nevertheless, there is no evidence of basal stem cells’ involvement in the infection, considering that basal cells are responsible for producing new neurons or sustentacular cells; this may partly explain why only a tiny proportion of COVID-19 patients develop chronic long-term anosmia [50,52].

However, in their report, Zazhytska et al. [45] claimed that the human olfactory epithelium virus burden is smaller and not entirely comparable with hamster models; an alternative mechanism for COVID-19-associated OD was presented and featured the ability to impair olfactory function without contaminating the epithelium or the sustentacular cells. By altering neuronal nuclear architecture, the virus also alters the genomic compartments that contain the genes for olfactory receptors and associated communication channels.

Regarding OD Pathophysiology of COVID-19, it has been described that the COVID-19 variations with the spike protein D614G mutation may facilitate membrane fusion, promoting infection of sustentacular cells. As a result, COVID-19 variants with the D614G mutation can be more likely to cause anosmia [53].

Another significant potential cause for COVID-19-associated OD is a systemic or local inflammatory process. Multiple cytokines, including TNF-α and IL-6, may cause damage to the olfactory epithelium directly or interfere with the cell signaling sequence and cause OD. Higher TNF-α levels were observed in the olfactory epithelium of SARS-CoV-2-infected individuals [54], and IL-6 blood levels were reported to be correlated with the presence of OD [55] (Figure 1).

Additionally, the cytokine response to COVID-19 infection may generate and mimic an organ-oriented autoimmune response in selected patients and trigger systemic or local autoimmune and autoinflammatory dysregulation [56]. In their published study, Ehrenfeld et al. [57] established an association between COVID-19 infection and the development of several autoimmune illnesses, including Immune thrombocytopenic purpura, Guillian–Barré syndrome, Miller–Fisher syndrome, and antiphospholipid syndrome. Furthermore, many autoimmune diseases, such as Systemic Lupus Erythematosus (SLE), myasthenia gravis, and systemic sclerosis, have been associated with numerous olfactory alterations mentioned above. Given the correlation between OD, autoimmune diseases, and dysregulated immune response in some COVID-19 patients, OD may be caused, at least to some extent, by autoimmune processes [58].

### 3.2. Omicron Variant: Impact on Olfactory Function

To accurately assess the rate and extent of anosmia prevalence in Omicron infections compared with other variants, it is imperative to evaluate the routes mentioned above for COVID-19-induced OD, including host cell invasion, the inflammatory process, and vaccination status.

It is worth noting that the Omicron variant has similar ACE2 receptor affinity as other variants and shares the D614G mutation associated with enhanced viral infection [38]. Although, in contrast to previously described SARS-CoV-2 variations such as SARS-CoV-2 614G, Gamma, and Delta, the Omicron variant infection, as simulated in hamster models, was shown to only slightly alter the olfactory epithelium’s pathology [59].

A crucial step in viral infection is viral membrane fusion, a process by which enveloped viruses enter host cells. The membrane fusion may occur through two distinct pathways: plasma cell membrane fusion or in endosomes after virus absorption by endocytosis [60]. Most commonly in COVID-19 variants, virions attach to the TMPRSS2 on the membrane’s surface to enter host cells. On the contrary, the Omicron variant favors the endosomal route, presumably due to less requirement of TMPRSS2 activation. Considering that sustentacular cells produce TMPRSS2 at high rates, the Omicron invasion of these cells is less prominent, which may partly explain why anosmia happens less often in Omicron variant infection [38].

In their literature review, Rodriguez-Selliva et al. [52] noted the Omicron variant can penetrate host cells by both pathways, demonstrates a broader range of target cells, and reproduces faster than other variants. However, the formation of a syncytium is diminished by the less reliant TMPRSS2 activation pathway. While other viral infections, including HSV-1, HIV, RSV, or SARS-CoV-2 variants, may lead to the development of a syncytium, and animal models have linked the phenomenon to the intensity of the symptoms, the reduced ability of the omicron variant to form a syncytium may lead to a lesser symptomatic infection.

An additional proposed mechanism of viral infection is the penetration ability of the olfactory epithelium mucus layer that acts as a biophysical barrier. While hydrophilic and acidic surface proteins are regarded as more soluble and suitable for mucus layer penetration, the Omicron variant mutations result in more hydrophobic and alkaline surface proteins and impair the infection of the epithelial cells [38].

The Omicron variant is believed to result in a less severe inflammatory reaction, a minor cytokine storm [52], and a subsequent diminished immunological reaction, leading to a lower incidence of olfactory impairment. In a hamster model, Bauer et al. [61] investigated and described the differences between neuronal cell invasion and inflammation among SARS-CoV-2 variations. In their report, hamsters infected with the Omicron variant had fewer inflammatory lesions within the olfactory mucosa than the Delta and D614G variants. However, the exact reason and process for why the Omicron variation inflects a less potent inflammatory response is yet to be well-known.

Boscolo-Rizzo et al. [39] published a study on patient and infection parameters of the Omicron variant predominant era compared with other eras. Their results found that smell or taste impairment incidence was significantly lower during the Omicron era. Enhanced vaccination status during the Omicron phase is one theory explaining the discrepancy, although not adequately proven.

First, no significant difference was found in anosmia prevalence according to vaccination status in their cohort analysis. Secondly, vaccination effectiveness is predominantly generated by developing IgG antibodies or cytotoxic T cells, which are not as effective as IgA in the mucosal immune response. Thirdly, the vaccine was considered less effective against the Omicron variation. Therefore, it cannot be reasonably assumed that previous vaccination status protects against olfactory system symptoms. Finally, vaccinated patients in the Delta variant dominant era suffered from olfactory alterations as a prevalent symptom. In conclusion, it appears that patients’ immune status does not significantly alter Omicron infection’s olfactory symptoms.

## 4. Conclusions

The olfactory system is intricate, complex, and imperative to everyday life. A loss or impairment in the olfactory system may lead to reduced ability to smell and taste and significantly harm the quality of life. As noted, olfactory functioning can be altered in multiple systemic, autoimmune, and autoinflammatory disorders, such as Sjögren syndrome, systemic lupus erythematosus, rheumatoid arthritis, autoimmune thyroiditis, and systemic sclerosis.

Along with autoimmune and systemic conditions, OD is a common symptom of COVID-19 infection that may appear as a presenting or a solitary symptom. Unexpectedly, the SARS-CoV-2 Omicron variant is far less associated with OD. Several hypotheses discussed in this review were proposed to elucidate this phenomenon. The absence of smelling impairment in Omicron-infected patients makes it more difficult to clinically suspect COVID-19 infection, which might delay diagnosis and isolation efforts. Although not as common as other variants, Omicron-related OD should remain a focus of further research. Olfactory issues were found to affect 10–30% of individuals throughout the Omicron-dominant era making this manifestation a public health concern [39].

## Figures and Tables

**Figure 1 diagnostics-13-00641-f001:**
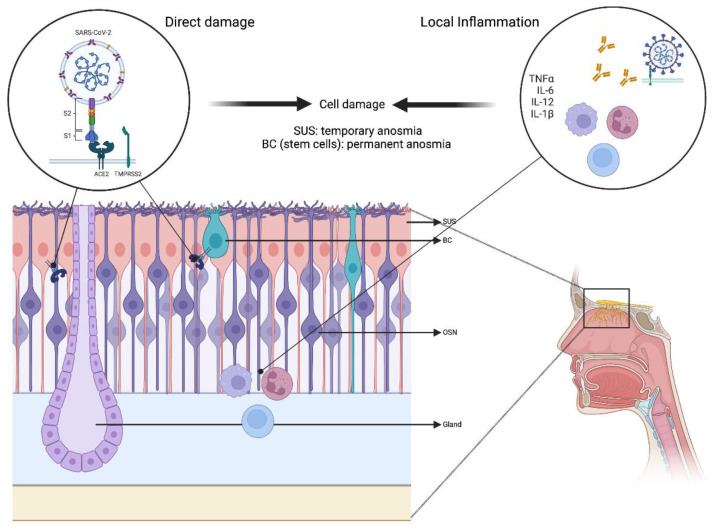
Pathophysiology of COVID-19-induced olfactory dysfunction. Distinct cell types make up the olfactory epithelium including OSN, SUS cells, and basal stem cells. The ACE2 receptor and the TMPRSS2 are crucial mediators of the COVID-19 virus’s ability to invade and fuse with host cells. ACE2 receptors are expressed on SUS and basal cells of the epithelium. Damage of the SUS cells putatively results in impairment of olfactory function. Given the integral role of basal cells in the regeneration and differentiation of neurons and sustentacular cells, basal cell damage can explain the persistent long-term anosmia amongst COVID-19 patients. Further, the olfactory function may be impaired by local or systemic inflammation resulting from the production of pro-inflammatory cytokines such as IL-6 and TNF-α. This inflammation may interfere with cellular signaling processes or directly damage the olfactory epithelium, leading to a disruption in the sense of smell. Abbreviations: SUS: sustentacular cells; BC: basal cells; TNF-α: tumor necrosis factor alpha; IL-6: interleukin 6; IL-12: interleukin 12; IL-1β: interleukin-1 beta; ACE2: angiotensin-converting enzyme 2; TMPRSS2: transmembrane serine protease 2; OSN: olfactory sensory neurons.

**Table 1 diagnostics-13-00641-t001:** Evidence of olfactory dysfunction in different autoimmune conditions.

Autoimmune/Autoinflammatory Condition	Evidence of Olfactory Dysfunction
Systemic lupus erythematosus (SLE)	Cohort studies showed that SLE patients have a decrease olfactory function when compared with controls, especially among those with psychiatric symptoms. The mechanism is to be elucidated. However, in animal models anti-P ribosomal antibodies intraventricular injection led to impaired smell, so this autoantibody may be involved. A cohort study has also shown association between anti-P levels and the OD. It can also be associated to the drug therapy, such as hydroxychloroquine.
Sjögren syndrome (SjS)	In various cohort studies, SjS patients have shown to have an impaired olfaction compared with controls, in particular among those with severe dry eyes. It was suggested that the hyposmia can be related to nasal mucosa dryness, but nasal septal perforation was also seen in this population. Interestingly, a decrease olfactory function was shown to be more frequent among those patients with anti-Ro and anti-La positivity.
Rheumatoid arthritis (RA)	Both gustatory and olfactory senses were shown to be decreased in RA patients. Olfactory bulbs of RA patients were shown to have a lower volume than healthy controls and a direct relationship between the volume and the olfactory function. Interestingly, the treatment with biological agents appears to be protective of this dysfunction, suggesting an inflammatory pathogenesis.
Systemic sclerosis (SS)	A cohort study shown that SS patients had abnormal olfactory test, with decreased function when compared with controls. The ability to discriminate between different odors were also shown to be impaired in SS patients.
Autoimmune thyroiditis	Thyroid hypofunction has been shown to lead to hyposmia in animal models. In fact, hormone replacement was shown to ameliorate the symptom.

## Data Availability

Not applicable.

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
