# Peer review of "Autoimmunity, COVID-19 Omicron Variant, and Olfactory Dysfunction: A Literature Review"

_diagnostics, 2023, doi:10.3390/diagnostics13040641_

Round 1
Reviewer 1 Report
The authors prepared a literature review on the topic linking COVID-19 omicron variant, olfactory dysfunction (OD) and autoimmune phenomena.
There are numerous meta-analyses on COVID-19 and OD.
The article is very long, often verbose in discussing known facts and at times less attention is paid to the topic.
In my opinion, the authors could write a systematic review according to PRISMA guidelines.
Fibromyalgia is not an autoimmune disease so I do not understand why authors would even discuss it here.
Lines 50-58 are direct copy-paste with 41-49 except for the lack of references in 50-58.
Lines 59-65: whole paragraph with not a single reference
Reference 8 is from 1989 - surely there has been something more recent on the topic.
Sjogren was a Swedish doctor and deserve a proper spelling throughout the text: Sjögren
Line 239: What does "As mentioned above" relate to?
Lines 250-252 - The reference is from AUG 2020, how can it relate to current situation (2022/2023)? The authors should separate what they claim from what others wrote.
Lines 409-410- I think most medically qualified personnel does not need a definition of syncytium from 1936.
The references do not use MDPI standard and in some cases even lack first authors REF 16 and 17 (?).
Reviewer 2 Report
Introduction: a refuse to cancel is present from line 50 to 55 page 2. imput error on line 63 (specific) and 64 (structure).I suggest to implement the paragraph on olfactory involvement in Covid infection (from line 89).
Sentence from line to 101 is not clear, it could be reproposed in an alternative way. Bulbectomy in animal models are nou useful to the aim of the study, please consider to cancel it. Table1 is interesting but the column with reference is not necessary (add references in the correct way in the text), it could be used to summarize data from the study to lighten the central one.
Round 2
Reviewer 1 Report
The authors have made changes to the article in line with my recommendations except for bibliography. This looks like no reference manager was used and certainly is not in MDPI references style. Please refer to MDPI guidelines to correct that.
In addition, there are numerous spelling mistakes and lack of spaces between words so please double check that while editing.
"References should be described as follows, depending on the type of work:
- Journal Articles:
1. Author 1, A.B.; Author 2, C.D. Title of the article. Abbreviated Journal Name Year, Volume, page range. - Books and Book Chapters:
2. Author 1, A.; Author 2, B. Book Title, 3rd ed.; Publisher: Publisher Location, Country, Year; pp. 154–196.
3. Author 1, A.; Author 2, B. Title of the chapter. In Book Title, 2nd ed.; Editor 1, A., Editor 2, B., Eds.; Publisher: Publisher Location, Country, Year; Volume 3, pp. 154–196. - Unpublished materials intended for publication:
4. Author 1, A.B.; Author 2, C. Title of Unpublished Work (optional). Correspondence Affiliation, City, State, Country. year, status(manuscript in preparation; to be submitted).
5. Author 1, A.B.; Author 2, C. Title of Unpublished Work. Abbreviated Journal Name year, phrase indicating stage of publication(submitted; accepted; in press)."